# Inflammatory, Metabolic, and Coagulation Effects on Medial Arterial Calcification in Patients with Peripheral Arterial Disease

**DOI:** 10.3390/ijms24043132

**Published:** 2023-02-05

**Authors:** Jovana Nikolajević, Mišo Šabovič

**Affiliations:** 1Department of Vascular Diseases, University Medical Centre Ljubljana, 1000 Ljubljana, Slovenia; 2Faculty of Medicine, Department of Internal Medicine, University of Ljubljana, 1000 Ljubljana, Slovenia

**Keywords:** vascular calcification, medial arterial calcification, hyperphosphatemia, hyperglycemia, inflammation, coagulation factors

## Abstract

Calcium deposits in the vessel wall in the form of hydroxyapatite can accumulate in the intimal layer, as in atherosclerotic plaque, but also in the medial layer, as in medial arterial calcification (MAC) or medial Möenckeberg sclerosis. Once considered a passive, degenerative process, MAC has recently been shown to be an active process with a complex but tightly regulated pathophysiology. Atherosclerosis and MAC represent distinct clinical entities that correlate in different ways with conventional cardiovascular risk factors. As both entities coexist in the vast majority of patients, it is difficult to estimate the relative contribution of specific risk factors to their development. MAC is strongly associated with age, diabetes mellitus, and chronic kidney disease. Given the complexity of MAC pathophysiology, it is expected that a variety of different factors and signaling pathways may be involved in the development and progression of the disease. In this article, we focus on metabolic factors, primarily hyperphosphatemia and hyperglycemia, and a wide range of possible mechanisms by which they might contribute to the development and progression of MAC. In addition, we provide insight into possible mechanisms by which inflammatory and coagulation factors are involved in vascular calcification processes. A better understanding of the complexity of MAC and the mechanisms involved in its development is essential for the development of potential preventive and therapeutic strategies.

## 1. Introduction

An accumulation of calcium deposits in the vessel wall, in the form of hydroxyapatite with a high degree of crystallization, is one of the first recognized features of atherosclerosis. Despite the accumulation of hydroxyapatite in the intimal layer, as in atherosclerotic plaque, calcium deposits may also accumulate in the medial layer, as in the development of medial calcifications. This clinical entity was first described by Möenckeberg in 1903 and is referred to as MAC or Möenckeberg medial sclerosis.

The accumulation of calcium deposits is a gradual process that begins with small, patchy calcifications located both extracellularly between elastic fibers in the media and intracellularly in vascular smooth muscle cells (VSMCs). As the disease progresses, small calcifications become larger and begin to fuse, leading to the development of linear or annular deposits that occupy an increasingly larger circumference of the vessel wall. In advanced stages of the disease, the entire circumference of the vessel wall may be affected, with calcium deposits filling the space between the elastic fibers and increasingly disfiguring the normal architecture of the media [1]. 

MAC can affect arteries of any type and size: large arteries of the elastic type (ascending aorta), medium-sized visceral and renal arteries, and small arteries with a diameter of at least 0.5 mm (coronary, temporal, uterine, ovarian, parathyroid, mammary, etc.) [1]. MAC is associated with older age, chronic kidney disease (CKD), and diabetes mellitus (DM) [2,3]. Nevertheless, MAC is also observed in patients with hyperparathyroidism, vitamin D disorders, vitamin K deficiency and anticoagulation therapy with vitamin K antagonists, osteoporosis, Kawasaki disease, rheumatoid arthritis, pseudoxanthoma elasticum, β-thalassemia, and many others [1]. 

Vascular calcification has long been considered a passive, degenerative process representing an end stage of atherosclerotic plaque development. Recent studies focusing on the risk factors and pathophysiology of mineral deposition in the vessel wall provide insight into the complex pathophysiology and show that, contrary to common belief, vascular calcification is an active and tightly regulated process.

Despite some similarities, atherosclerosis and MAC are now considered distinct clinical entities with different risk factors, pathophysiology, clinical course, and consequences. As both entities coexist in the vast majority of patients with peripheral arterial disease (PAD), it is quite difficult to estimate the relative contribution of specific risk factors to their development. The development of atherosclerotic plaques is the result of subendothelial lipid deposition, macrophage accumulation, smooth muscle cell proliferation, and dysfunctional accumulation and reorganization of extracellular matrix proteins in response to chronic arterial inflammation [4]. In contrast, the pathophysiology of MAC is not fully understood. According to the available evidence, the pathophysiological mechanisms that play an important role in the development of arterial calcification are numerous and distinctly different from those of atherosclerosis. In this article, we will focus on the role of inflammatory, metabolic, and coagulation processes in the pathophysiology of MAC in patients with PAD.

## 2. Clinical Relevance of MAC in Patients with PAD

The clinical consequences of MAC are not limited to the vessel wall of the affected vessel; its effect is rather systematic and spread across the cardiovascular (CV) system. The accumulation of calcium deposits in the form of concentric layers between the elastic fibers in the media leads to stiffening of the arteries and hemodynamic changes. This is best illustrated by an increase in pulse wave velocity and pulse pressure, as well as pulse wave deformation [5,6]. Changes in blood flow pulsatility are associated with changes in tissue and organ perfusion, leading to microvascular dysfunction, as suggested by a marked decrease in mean and diastolic arterial pressures following vasodilatation [7]. In advanced stages of the disease, the loss of arterial wall elasticity is so profound that there is a functional loss of pulsatile blood flow [1]. Further on, arterial stiffening is associated with increased cardiac afterload, leading to cardiac hypertrophy and diastolic dysfunction [8,9]. Finally, arterial stiffness is a strong predictor of future CV events and all-cause mortality [10]. 

As mentioned earlier, atherosclerosis and MAC occur together in the vast majority of patients with PAD, so there is an important overlap between the two clinical entities. In general, MAC does not cause arterial lumen narrowing but could promote the development of atherosclerotic plaques by inducing subendothelial hyperplasia [1]. Despite that, MAC could also affect atherosclerosis by limiting positive vascular remodeling and promoting a negative one [11]. Therefore, the clinical manifestations of PAD are more severe and progress more rapidly when atherosclerotic plaques and MAC coexist in the same arterial segment.

The ankle brachil index (ABI) has important limitations in patients with arterial stiffness, such as those with diabetes or chronic kidney disease [12]. Arterial stiffening results in the overestimation of ABI. The toe–brachial pressure index (TBPI) is recommended as an alternative to the ABI, but its reliability and added benefit compared with the ABI are scant [13]. The American Heart Association (AHA) and European Society of Cardiology/European Society for Vascular Surgery (ESC/ESVS) guidelines recommend Doppler waveform analysis as a useful tool for the diagnosis of PAD when arterial stiffening is present [14,15]. 

There is extensive evidence supporting the relationship between Doppler waveform contours with the presence and extent of PAD. The contours of the Doppler waveform change with PAD progression, from normal, triphasic waveforms towards biphasic or monophasic ones in the case of high degree stenosis [16]. 

MAC also limits the success of percutaneous revascularization procedures on peripheral arteries. The available data show that endovascular procedures in patients with MAC are associated with a higher complication rate (dissection, suboptimal stent deployment, and decreased drug absorption when drug-eluting balloons are used) and a greater risk of restenosis, as well as severely limiting the options and success of surgical revascularization [17,18,19]. As a result, the success rate of revascularization procedures is lower in patients with MAC and critical limb ischemia (CLI), and improvement is often short-term, resulting in a higher limb amputation rate [17,20]. 

Not only the presence but also the extension of MAC, expressed as the proportion of vessel circumference affected by calcification of the medial layer, is associated with a poor prognosis in patients with PAD. Patients with CLI are more likely to have extensive calcification of the media, often encompassing the whole vessel circumference [21]. Extensive MAC is independently associated with an increased 10-year mortality risk and risk of limb amputation [22]. The extension of medial calcification is associated with an increased mortality risk, irrespective of the revascularization method used [23]. A meta-analysis, including more than 6500 patients, showed that the presence of MAC on below-knee arteries is associated with a more than two-fold greater risk of lower limb amputation [24]. 

## 3. Cornerstones of MAC Pathophysiology

Nowadays, arterial calcification is considered to be an active and tightly regulated process, the main features of which are similar to the process of intramembranous bone formation and odontogenesis [25]. The most important step in MAC pathophysiology is the molecular reprogramming, followed by the phenotypic and functional transformation of VSMCs from a contractile to an osteochondrogenic phenotype. In addition, the current hypothesis recognizes the importance of other processes observed during the development and progression of MAC in the vessel wall: VSMC apoptosis and formation of matrix vesicles, inflammation, disruption of calcium/phosphate homeostasis, and degradation of the extracellular matrix.

VSMC phenotypic transformation is characterized by the loss of contractile markers and increased expression of bone-related genes: *bone morphogenetic protein (BMP), bone sialoprotein, osteocalcin, Msh homeobox 2 (Msx2), transcription factors Cbfa1/Runx2, and Sox9*, etc. [26,27]. It is still not clear which factors are responsible for triggering and the initiation of phenotypic change. Recent studies have reported that the phenotypic change could result from VSMC senescence, as the senescent phenotype has been associated with the upregulation of molecules that promote osteogenic differentiation [28]. It was proposed that senescent VSMCs, as paracrine sources, might promote osteogenic and inflammatory responses while modulating calcification locally in the vessel wall [29]. Furthermore, it was suggested that senescent VSMCs might promote the development and progression of atherosclerotic plaques by inducing subendothelial hyperplasia [1].

One of the most important morphological features of VSMC phenotypic change is the formation of matrix vesicles, also observed in chondrocytes, osteoblasts, and odontoblasts. Matrix vesicles are formed by budding from the plasma membrane and contain calcium-binding proteins and phosphatases required for the formation of the nidus and initiation of mineralization [30]. Matrix vesicles could represent apoptotic bodies, the remnants of apoptotic cells, as the BAX protein, a pro-apoptotic member of the bcl-2 family, was found in them [31]. 

Aging itself is accompanied by an accumulation of age-related elastin changes: mechanical breakage, proteolysis and calcification, and glycation or lipid peroxidation of elastin fibers due to the very low elastin turnover rate [32]. All these changes in elastin structure and function could trigger the inflammatory process and activation of matrix metalloproteinases (MMPs), thus promoting extracellular matrix (ECM) degradation, which is considered to be an underlying mechanism for a wide variety of vascular diseases [32]. Matrix metalloproteinases play a critical role in elastin degradation, ECM disorganization, and subsequent calcification. In vitro studies have shown that MMP knockout mice are resistant to elastin degradation and calcification after aortic injury [33]. The active form of MMP-2 is also found in matrix vesicles derived from VSMCs. It has been reported that the inhibition of MMP-2 activity in matrix vesicles prevents VSMC calcification [34]. However, a study in the uremic mouse model showed that elastin degradation itself is not sufficient to induce the development of MAC without the phenotypic switch and apoptosis of VSMCs [35]. 

The primary purpose of the following section is to illustrate the complexity of MAC pathophysiology by detailing the mechanisms by which metabolic, inflammatory, and coagulation factors are involved in the process of vascular calcification in PAD (Figure 1). We also aim to provide insight into the broad spectrum of factors, the molecular pathways, and their interplay that could modulate MAC pathophysiology to highlight opportunities in the search for potential new preventive and therapeutic targets. The inflammatory, metabolic, and coagulative factors involved in MAC pathophysiology and possible mechanisms by which they contribute to MAC development and progression are listed in Table 1.

### 3.1. Inflammatory Factors Associated with MAC

As both CKD and DM are considered inflammatory conditions, inflammatory signaling was proposed to be a key factor in the high prevalence of MAC observed in these patient groups [66]. 

It is widely accepted that atherosclerosis is an inflammatory disease as low level chronic inflammation is one of the crucial risk factors for the development and progression of atherosclerotic plaques. Inflammation modulates some of the most important steps in the development of atherosclerotic plaques: the initial recruitment of circulating inflammatory cells, the formation and progression of plaques, and, eventually, rupture of the atherosclerotic plaque, leading to thrombotic complications and acute ischemic events [67]. On the contrary, MAC is not considered predominantly an inflammatory disease. However, recent evidence has suggested that the role of inflammation in MAC pathophysiology has probably been underestimated. Inflammation contributes to the development and progression of media calcification indirectly, by modulating the signaling pathways involved in its pathogenesis. 

Probably the most important cytokine in MAC pathophysiology is tumor necrosis factor-α (TNF-α), involved in both vascular and bone physiology. Macrophages are the main source of TNF-α synthesis, after they are activated by infection, oxidized LDL, or the ECM breakdown products (e.g., fibronectin or laminin glycoprotein components) [68,69]. An in vitro study reported that TNF-α modulates the expression of genes involved in VSMC transformation to osteoblast-like cells, at least partially, by the activation of the cAMP pathway. Furthermore, it was reported that TNF-α modulates the expression of genes involved in ECM formation and mineralization [36]. The same study revealed that TNF-α has a reciprocal effect on VSMCs in the vessel wall and osteoblasts in bones: it stimulates the osteochondrogenic differentiation of VSMCs and inhibits osteoblast differentiation in bone. This is probably due to the activation of different signaling pathways in the vessel wall and bones, suggesting that inflammation may represent the link between osteoporosis and vascular calcification that commonly coexist in the elderly. Its role in the regulation of VSMC and osteoblast proliferation and differentiation is probably modulated via the runt-related transcription factor RUNX2 [70]. 

Some cytokines could also modulate other important pathways involved in MAC pathophysiology. For example, IL-1β, also secreted by activated macrophages, is involved in the regulation of phosphate metabolism. IL-1 β increases the expression of tissue-nonspecific alkaline phosphatase (TNAP) in VSMCs, independently of RUNX2, thus decreasing the level of pyrophosphate ions in extracellular spaces and promoting vascular calcification [37]. 

A better understanding of the mechanisms behind cellular senescence led to the hypothesis that the development of a senescence-associated secretory phenotype (SASP) could be the key step that triggers the phenotypic transformation of VSMCs. The development of SASP is accompanied by the production of growth factors, proteases, and inflammatory cytokines, with both osteo-inductive and pro-inflammatory characteristics: bone morphogenetic protein (Bmp-2), Il-1β, and Il-6 [29,38,39]. This could be the missing link underlying the association observed between vascular calcification and older age. An in vitro study reported the overexpression of prelamin A, involved in DNA damage repair signaling, in senescent VSMCs. Prelamin A overexpression is associated with increased mRNA levels of BMP2, OPG, and IL-6, followed by osteogenic differentiation and VSMC mineralization [29].

### 3.2. Metabolic Factors Associated with MAC 

The strong association of MAC, observed with both CKD and diabetes, guided an extensive research work, aimed at better understanding the mechanisms by which metabolic disturbances in CKD and DM contribute to vascular mineralization. Considering their relative importance for the initiation and progression of MAC and the amount of data available on the topic, we decided to limit the scope of this section to hyperphosphatemia and hyperglycemia. However, by including the results from recent studies, revealing potential new signaling pathways that could interfere with MAC pathophysiology, we illustrate the complexity of the process and the numerous molecules and pathways that could serve as targets for potential new preventive or therapeutic strategies. 

#### 3.2.1. The Role of Hyperphosphatemia in Vascular Calcification

Hyperphosphatemia is considered to be one of the most important factors in the development of vascular calcification. Initially, it was hypothesized that a high phosphate concentration drives vascular calcification simply by causing the precipitation of calcium phosphate once the calcium phosphate solubility product is exceeded. Under physiological circumstances, serum levels of calcium and phosphate are quite close to those at which the precipitation of calcium phosphate occurs, yet spontaneous precipitation is counteracted by the action of various proteins that serve as precipitation inhibitors [40]. That being said, in patients without hyperphosphatemia, either the loss of precipitation inhibitors or their inadequate functioning could lead to calcium phosphate precipitation. 

The strongest endogenous inhibitor of mineralization is inorganic pyrophosphate, but other circulating or locally synthesized proteins could also serve as potent calcification inhibitors. Pyrophosphate is mainly produced during the extracellular hydrolysis of ATP by the enzyme ecto-nucleotide pyrophosphatase/phosphodiesterase (eNPP) [71]. ATP is also a direct inhibitor of calcification and is generated intracellularly, probably in mitochondria, using adenosine from the extracellular space [72]. Adenosine, a by-product of pyrophosphate generation, is transported across the plasma membrane to mitochondria by nucleoside transporter 1 (ENT1, Slc29a1). Loss of this transporter in mice is associated with ectopic calcification of paraspinal tissues [73]. Studies in a mouse model of premature aging showed that impaired synthesis of intracellular ATP, due to mitochondrial dysfunction, is associated with a reduction in extracellular pyrophosphate concentration as well as vascular calcification development [74]. 

Fetuin A, a circulating protein, plays an important role in calcification inhibition by directly binding to hydroxyapatite crystals, thus inhibiting their growth. Fetuin-A knockout-mice spontaneously develop soft tissue calcification of the heart, vessels, kidney, testis, and skin [75]. However, serum fetuin-A levels did not correlate with the presence of vascular calcifications in patients on hemodialysis [41,76]. VSMCs produce several other inhibitory proteins, such as matrix-Gla protein (MGP), osteopontin, and osteoprotegerin (OPG), which are excreted to the ECM through extracellular vesicles to prevent vascular mineralization [40]. As a result of extensive research work and a better understanding of MAC pathophysiology, other molecules, indirectly involved in phosphate homeostasis (e.g., klotho/fibroblast growth factor 23 (FGF23) and vitamin K) have recently emerged as potential inhibitors of calcification [40,77]. 

Phosphate transport into cells is primarily mediated by sodium-dependent phosphate cotransporters. Only the expression of type III sodium-dependent phosphate cotransporters (Pit-1 and Pit-2) was detected in VSMCs, with Pit-1 being more abundant than Pit-2 [78,79,80]. Competitive inhibition of phosphate transporter results in the dose-dependent inhibition of phosphate uptake, thus confirming their crucial role in intracellular phosphate metabolism. In vitro studies have reported that phosphate transporters are saturated at the physiological phosphate serum level, leading to the hypothesis that the increase in serum phosphate level is not followed by an immediate increase in the intracellular phosphate load [81]. The increased concentration of intracellular phosphate results from the upregulation of the sodium-dependent phosphate cotransporter Pit-1 [80]. Moreover, this step represents a crossroads for different signaling pathways involved in MAC pathophysiology, as the upregulation of phosphate transporters could be triggered by conditions other than CKD. For example, Pit-2 expression could be increased by H^+^ ions instead of Na^+^ at acidic pH (6.0), but this pathway is probably of small clinical relevance [80]. Platelet-derived growth factor (PDGF) is also reported to increase Pit-1 expression in VSMCs in vitro [82]. 

High serum phosphate levels could contribute to vascular calcification not only by promoting the precipitation of hydroxyapatite crystals but also by modulation of some of the most important signaling pathways in MAC pathophysiology: the phenotypic transformation of VSMCs to osteoblast-like cells, extracellular matrix remodeling, VSMC apoptosis, inhibition of monocyte/macrophage differentiation into osteoclast-like cells, and FGF23 levels and Klotho expression [40,83]. 

Almost all the main events in the pathophysiology of MAC are at least partially modulated through *CBFA1/RUNX2* (core-binding factor subunit 1α/runt-related transcription factor 2) expression. In vitro exposure of VSMCs to elevated phosphate concentrations resulted in phenotypic transition from the contractile to synthetic phenotype, as a result of concomitant downregulation of contractile proteins (e.g., smooth muscle α-actin (Acta2), SM22α (Tagln), smooth muscle cell myosin heavy chain (Myh11), and the upregulation of *CBFA1/RUNX2* expression [42]. *Cbfa1* obviously plays a critical role in the process of mineralization, as it was reported that *Cbfa1* knockout mice fail to form mineralized bone [43]. *Cbfa-1* is an osteoblast-specific transcription factor involved in the process of bone mineralization by regulating osteoblast differentiation and increasing the expression of bone matrix components (collagen type I, osteocalcin, and osteopontin) [27,44,45,46]. Increased expression of *CBFA1/RUNX2*, followed by an increased expression of osteopontin and type I collagen, was reported in both the intima and media layers of calcified inferior epigastric arteries from dialysis patients [47]. *RUNX2* is also involved in the regulation of osteoblast proliferation and apoptosis through the regulation of TNFα signaling [70]. Finally, *RUNX2* is reported to modulate the expression of inflammatory cytokines that accelerate macrophage infiltration and phenotypic changes as well as the differentiation of vascular osteoclasts [48]. 

Hyperphosphatemia induces extracellular matrix remodeling by increasing matrix metalloproteinases and cysteine protease expression, resulting in increased degradation of matrix proteins and the generation of bioactive elastin peptides as well as increased collagen synthesis [49]. In addition, hyperphosphatemia induces the expression of enzymes that regulate collagen crosslinking and organization [50]. 

Increased serum phosphate levels could induce apoptosis in terminally differentiated chondrocytes [84]. The mechanism is still not completely understood, but it is assumed that an increased serum phosphate level, followed by an increased intracellular phosphate level, induces apoptosis by disrupting the normal mitochondrial energy metabolism [85]. A high serum phosphate level is reported to induce apoptosis and subsequent calcification in aortic smooth muscle cells in vitro in a time- and dose-dependent manner [86]. On the other hand, statins could block phosphate-induced calcification in human aortic smooth muscle cells, but this effect is probably not mediated by the modulation of sodium-dependent phosphate cotransporters but the restoration of the survival pathway mediated by growth arrest-specific protein 6 (Gas6). The restoration is not dependent on the transcription rate but is probably mediated by the inhibition of Gas6 mRNA degradation [86]. 

The process of bone formation requires a delicate balance between mineral deposition by osteoblasts and mineral resorption by osteoclasts. Thus, it was hypothesized that vascular calcification could result from a disrupted balance of mineral deposition and resorption in the vessel wall. VSMCs are not the only cells that undergo phenotypic changes induced by increased serum phosphate levels, as monocyte/macrophages are also able to transform into osteoclast-like cells [51]. After being activated, macrophages transform from the M1 type to an alternative type, sharing a lot of markers and biological functions with the M2 macrophage type. Those two cell types have distinct physiological characteristics: classical macrophages (M1) promote inflammation and inhibit cell proliferation, resulting in tissue damage. Conversely, the M2 type promotes cell proliferation and causes tissue repair [52]. The M1 type induces TNAP expression, while the M2 type induces ectonucleoside triphosphate diphosphohydrolase 1 (eNPP1) expression in VSMCs. As a result, an increased release of extracellular ATP and pyrophosphate was observed in the VSMCs/M2 co-culture, followed by a reduction in calcium phosphate deposition [52]. Furthermore, an increase in the serum phosphate level was reported to inhibit in vitro osteoclast monocyte/macrophage progenitor differentiation into osteoclast-like cells in a dose-dependent manner. This action is probably modulated via the down-regulation of RANKL-induced JNK Akt and NF-κB activation pathways [53]. 

FGF-23, a hormone secreted by osteocytes, is considered to be one of the key regulators of serum phosphate levels, vitamin D metabolism, and secondary hyperparathyroidism. The serum level of FGF-23 increases in patients with CKD leading to increased renal phosphate excretion and decreased absorption from the intestine [54]. Additionally, FGF-23 suppresses parathyroid hormone (PTH) secretion, *PTH* gene expression, and parathyroid cell proliferation [55]. Due to weak binding to its receptor, FGF-23 usually requires a co-factor, Klotho, except in the state of advanced CKD and excessive levels of FGF-23, where it exerts its biological action even without Klotho [56]. FGF receptors, but not Klotho receptors, were found on the VSMC membrane [57]. In vitro studies in CKD animals lacking Klotho reported an increased expression of the sodium-dependent phosphate cotransporter and *CBFA1/RUNX2* in VSMCs. It was also reported that the addition of Klotho results in reduced phosphate uptake in VSMCs due to the suppression of the sodium-dependent phosphate cotransporter and prevention of VSMC transformation to osteochondrogenic cells [58]. 

Hypercalcemia is also an important factor contributing to the development and progression of vascular calcification. Epidemiological studies reported a direct correlation between serum phosphate and the calcium phosphate product (Ca × Pi) and vascular calcification in patients with end-stage renal disease [87]. In vitro studies reported that vascular calcification depends on the activity of the Pit-1 transporter, whose expression could be increased in the setting of increased serum levels of calcium and phosphate, as measured by Pit-1 mRNA levels [88]. 

In patients with CKD, vascular calcification is associated with the duration of hemodialysis and taking of calcium-containing phosphate binders [3]. Compared to non-calcium-containing phosphate binders, the use of calcium-containing phosphate binders could result in hyperphosphatemia and hypercalcemia and is reported to be associated with progressive coronary and aortic calcification in patients on hemodialysis [89]. 

#### 3.2.2. The Role of Hyperglycemia in Vascular Calcification

Diabetes mellitus comprises a cluster of metabolic disturbances, each of which could contribute to vascular calcification. Due to interconnections between the different signaling pathways involved in the pathophysiology of its metabolic disturbances, it is to be expected that the effect of particular disturbances is additive. Thus, it is not surprising that MAC is more prevalent in patients with DM than in patients with CKD [2]. Of note, MAC prevalence is higher in patients with DM type 2 than DM type 1, supporting the hypothesis that, beyond glucose homeostasis, other metabolic disturbances that accompany DM type 2 also promote the calcification process. 

Four distinct mechanisms by which hyperglycemia could lead to vascular calcification were proposed: increased oxidative stress resulting from the activation of the polyol pathway, inflammatory response induced by advanced glycation end products (AGEs), a decreased level of nitric oxide and an increased level of endothelin-1, resulting from the activation of the protein kinase-C pathway, and finally increased levels of plasminogen activator inhibitor-1 and TGF-β resulting from the stimulation of the hexosamine pathway [90]. 

The principal mechanism, by which hyperglycemia drives vascular calcification, is thought to result from the accumulation of AGEs in the vascular wall. AGEs are formed by the nonenzymatic glycation of proteins, lipids, and nucleic acids. AGEs exhibit numerous harmful effects on the vascular wall: increased inflammation and oxidative stress, increased glycation of low-density and high-density lipoproteins (LDL, HDL), activation of the pro-inflammatory inducible nitric oxide (NO)-synthase (iNOS), and decreased NO availability [59]. An accumulation of AGEs is associated with increased synthesis of cytokines, e.g., insulin-like growth factor-1 (IGF-1) and PDGF, leading to the migration of monocytes and macrophages as well as the proliferation of VSMCs [60]. An accumulation of AGEs is particularly important for proteins with a long biological half-life, such as collagen and elastin, as its crosslinking is only neutralized by the synthesis of new protein [61]. Elastin–AGEs crosslinking results in increased binding of calcium ions to elastin and increased elastin stiffness [61]. The accumulation of AGEs in the aortic wall is associated with increased aortic stiffness in rats [62]. In vitro studies reported that AGEs could also induce osteogenic differentiation of VSMCs by activating the Notch/Msx2 signaling pathway [63]. 

The association between insulin resistance/DM and increased serum OPG levels is well documented [64]. OPG is a glycoprotein, a member of the TNF-α receptor family, playing a key role in bone turnover by actively regulating osteoclastogenesis and bone resorption [65]. OPG inhibits vascular calcification by competitive inhibition of the receptor activator NF-κB ligand (RANKL), thus inhibiting osteoclastogeneis and bone resorption [91]. Paradoxically, clinical studies consistently report an association between increased serum OPG levels with increased vascular calcification as well as coronary artery disease, stroke, and future CV events [92]. However, it is still unclear whether OPG is a marker or a mediator of vascular calcification and CV risk. OPG levels significantly correlate with insulin resistance, C-reactive protein (CRP), IL-1, PDGF, and TNF-α levels but do not correlate with common CV risk factors: hypertension, dyslipidemia, or abdominal obesity [64,91]. Consequently, it was proposed that OPG is a potential marker of inflammation, but it also exerts an ability to modulate alkaline–phosphatase activity as well as apoptotic and osteogenic responses [93]. 

A recent finding suggests that osteocalcin, the most abundant noncollagenous protein in the bone, is also involved in the regulation of energy metabolism, indicating the skeleton as potentially an endocrine organ. After synthesis, osteocalcin undergoes γ-carboxylation, which is a vitamin K-dependent process. A great amount of osteocalcin binds to bone hydroxyapatite, while a small fraction is released into circulation and exerts endocrine actions by regulating glucose metabolism [94]. In chondrocytes and VSMCs, osteocalcin stimulates glycolysis and simultaneously inhibits gluconeogenesis, by decreasing the expression of glucose-6-phosphatase and phosphoenolpyruvate carboxykinase enzymes [94]. An in vitro study has reported that osteocalcin-deficient mice were obese, hypoinsulinemic, and hyperglycemic. Adding the recombinant bone Gla protein (BGP) to wild-type mice resulted in improved glucose metabolism and reduced fat mass [94,95]. This finding supports the hypothesis that glucose metabolism, bone mineralization, and vascular calcification processes are interconnected and probably regulated by the same metabolic/endocrine axis.

### 3.3. Coagulation Effects on MAC 

To date, there is ample evidence of an association between the presence of calcium deposits in atherosclerotic plaques and an increased risk of arterial thrombosis. In contrast, the evidence on the association between calcium deposits in the media and an increased risk of arterial thrombosis in patients with PAD is scarce [96]. The reason for such a discrepancy could be that both processes, atherosclerosis and MAC, coexist in the vast majority of patients with PAD. Furthermore, the most widely used imaging modalities (computed tomography angiography and digital subtraction angiography) are not able to distinguish between intimal and medial calcium deposits, so it is impossible to estimate their particular contribution to the development of arterial thrombosis.

A histopathological study performed on 176 lower limb arteries from 60 patients with CLI who underwent amputation reported a 5.6% prevalence of thrombotic arterial occlusion. MAC without significant atherosclerotic plaque burden was detected in 70% of the specimens [97]. Another study on arterial specimens from 95 patients who underwent limb amputation reported occlusive thrombosis in the absence of significant atherosclerotic plaque in 67.5% of specimens. The same study showed that thrombosis resulting from atherosclerotic plaque rupture was more common above the knee, whereas MAC-related thrombosis was more common in the arteries below the knee [98]. Thus, the contribution of MAC to the development of arterial thrombosis is certainly underestimated and could be of even greater importance than atherosclerotic plaque rupture. 

The exact mechanism by which MAC leads to arterial thrombosis remains unknown. The stasis of blood, resulting from loss of pulsatile blood flow due to media calcification, is one of the key risk factors in Virchow’s triad and could increase the risk of arterial thrombosis. On the other hand, MAC could induce subendothelial hyperplasia of the intima, characterized by increased cellularity (e.g., myofibroblasts, fibroblasts, fibrocytes) and infiltrations of the intima, which could protrude into the lumen or lead to the development of ulcerations [1]. Finally, extensive research on the mechanism of arterial calcification recently revealed an overlap between complex mechanisms of MAC with coagulation processes. 

Vitamin K serves as a cofactor in the process of the carboxylation of glutamic acid residues, an important step in the synthesis and activation of so-called vitamin K-dependent proteins (VKDPs). This group comprises 17 different protein types identified in the bone, heart, and vessel wall, and is divided into two groups: hepatic (coagulant factors II, VII, IX, and X as well as anticoagulant proteins C, S, and Z) and extrahepatic MGP, growth arrest-specific protein 6 (Gas6), osteocalcin, Gla-rich protein (GRP), periostin (isoforms 1–4), periostin-like factor (PLF), proline-rich Gla protein (PRGP) 1, PRGP2, transmembrane Gla protein (TMG) 3, and TMG4) [61]. Due to a high content of γ-carboxy glutamic acid, both groups have a high affinity for Ca^2+^ ions [99]. 

Some of the proteins from the extrahepatic group play key roles in the pathophysiology of vascular calcification. Osteocalcin inhibits the osteoblastic differentiation of VSMCs, MGP modulates the inhibition of ectopic calcification, and Gla-rich protein (GRP) inhibits the expression of osteogenic genes and reduces the transformation of VSMCs into osteoblast-like cells, whereas Gas-6 affects VSMC apoptosis and the formation of the nidus for calcium phosphate precipitation and is also involved in inflammation and thrombogenesis [100,101]. VSMCs continuously produce MGP and Gas-6, which then undergo carboxylation. The γ-carboxylated VKDPs bind calcium ions, thereby inhibiting crystal growth in the extracellular matrix. Both normal and calcified vessels express MGP, but normal vessels have a predominance of carboxylated MGP, whereas uncarboxylated MGP predominates in calcified vessels [102]. It seems that it is the ratio of carboxylated to uncarboxylated protein that determines the biological activity of both MGP and Gas-6 [103]. This is also true for Gla-rich protein (GRP), whose inhibitory activity depends on carboxylation: undercarboxylated protein lacks calcification inhibitory capacity, thereby promoting vascular mineralization and osteochondrogenic differentiation through α-smooth muscle actin upregulation and osteopontin downregulation [104,105]. However, its anti-inflammatory activity seems to be independent of its carboxylation level [106]. Under physiological conditions, osteocalcin is not secreted by VSMCs, but its expression is detected in VSMCs undergoing osteochondrogenic transformation. Of note, only the carboxylated form of osteocalcin is able to drive metabolic changes and VSMC transformation [107]. 

It is thus evident that the biological activity of some the most important proteins involved in MAC pathophysiology is highly dependent on their carboxylation, the process that requires vitamin K as a cofactor. During carboxylation, vitamin K must be converted from an oxidized to a reduced form, a reaction catalyzed by vitamin K epoxide reductase (VKOR). In the process of carboxylation, vitamin K is recycled by returning to its oxidized state. Warfarin shares a common ring structure with vitamin K and interferes with VKOR, thus interrupting vitamin K recycling [108]. As a result, vitamin K deficiency, as well as therapy with vitamin K antagonists, limit carboxylation and thereby the activation of VKDPs, thereby interfering with both anticoagulation processes and bone and vascular mineralization. 

Due to the differences between hepatic and extra-hepatic vitamin K metabolism, vitamin K deficiency primarily affects extra-hepatic VKDPs carboxylation. In the liver, the recycling of inactive vitamin K could also be catalyzed by DT-diaphorase, whose activity is not affected by warfarin. The activity of this enzyme is extremely low in extra-hepatic tissues and VSMCs, thus making the carboxylation of extra-hepatic VKDPs more sensitive to warfarin treatment [109]. Studies on Wistar Kyoto rats reported that the simultaneous administration of warfarin and vitamin K does not affect the synthesis of coagulation factors but is associated with diffuse medial calcification that could be only partially relieved by a vitamin K-rich diet [110]. 

The prevalence of vitamin K deficiency in patients with chronic kidney disease (CKD) is estimated to be more than 60% [111]. A vitamin K Italian (VIKI) dialysis study reported vitamin K2 deficiency to be significantly associated with aortic and iliac arterial calcification [112]. Vitamin K supplementation has been reported to be associated with improvement in arterial stiffness among both healthy adults and patients with end-stage renal disease requiring hemodialysis [113]. Human studies have also reported an association between high vitamin K2 intake, reduced coronary artery calcification, and CVD risk [114]. 

Long-term warfarin use was reported to be associated with increased coronary and extra-coronary calcifications [115,116]. Studies on an animal model of CKD showed that a low therapeutic dose of warfarin promotes an accumulation of calcium deposits in the vessel wall. This treatment did not promote calcification in rats without kidney dysfunction, suggesting that vitamin K deficiency only potentiates the effect of other factors that promote vascular calcification in the setting of CKD [117]. A recent study reported a significant reduction in vitamin K deficiency, assessed by measuring levels of inactive forms of prothrombin and osteocalcin after 3 months of replacing warfarin with rivaroxaban. The study also reported a significant reduction in arterial stiffness assessed by using brachial–ankle pulse wave velocity (PWV) [118]. Warfarin treatment is also reported to be associated with an increased calcification of coronary arteries, thoracic aorta, aortic, and mitral valve, as well as with the structural and hemodynamic progression of aortic stenosis [119,120,121]. Warfarin treatment is also associated with increased atherosclerotic plaque inflammation, oxidative stress, calcification activity, and plaque progression compared to the novel oral anticoagulants (NOACs) treatment [122]. So far, there are no data on the association between warfarin treatment with inflammation or oxidative stress in MAC. 

Genetic polymorphism of the *VKOR* gene could be partially responsible for interindividual variation of the anticoagulant effects of warfarin [123]. As the process of vitamin K recycling is also essential for vascular mineralization, it can be expected that genetic polymorphisms of the *VKOR* gene could also affect susceptibility to vascular mineralization. However, further studies in this area are needed to elucidate this theory.

## 4. Conclusions and Future Perspectives

Although the pathophysiological mechanism of MAC is now much better understood than it was a few decades ago, several important issues related to MAC remain to be addressed. These are the circulating biomarkers, risk stratification, visualization methods, and finally prevention and therapeutic approaches. 

The value of numerous markers has already been tested. Most studies are of a poor quality, were mainly cross-sectional, and were influenced by confounding factors. The most recent meta-analysis included several biomarkers: calcium, phosphate, parathyroid hormone, vitamin D, pyrophosphate, OPG, RANKL, FGF-23, Klotho, osteopontin, osteocalcin, MGP and its inactive forms, and vitamin K. Of all the markers included, phosphate, osteopontin, and FGF-23 were found to be suitable biomarkers for MAC [124]. However, further studies are needed to test their predictive value for the development of MAC. 

Visualization and quantification are also important aspects in the assessment of MAC. The ability of noninvasive techniques to distinguish between atherosclerotic plaques and MAC and quantify MAC are limited. The best option is intravascular ultrasound (IVUS) or optical coherence tomography (OCT), but these methods are invasive and expensive and, therefore, not widely available. In the future, the utility of noninvasive methods could be improved by integrating new software that allows automatic, operator-independent quantification of calcium deposition in specific vessel areas and the estimation of the relative contribution of atherosclerosis/MAC to the total calcium burden.

Risk stratification models for the development and progression of MAC are not yet available. Apart from the association with DM and CKD, there are few data on the association with other biomarkers or risk factors that could potentially be included in a risk stratification model. The genetic predisposition to the development of MAC has not been fully elucidated nor has the importance of epigenetic changes and miRNA in the MAC pathophysiology.

Finally, despite a better understanding of the pathophysiology, there is still no effective preventive and therapeutic approach for MAC. Despite the fact that various drugs have been tested, to date, there is no evidence that MAC is reversible. Therefore, all current strategies aim to inhibit the progression of MAC or prevent the deposition of calcium in the vessel wall. As the pathophysiology is still not clearly understood, the current approach depends on the existing comorbidities and risk factors, which means that there is no specific but also no individualized treatment. Further studies to help elucidate the pathophysiology of MAC will likely reveal new preventive and therapeutic targets. Finally, there is increasing evidence of potential interactions between the various drugs used to treat patients’ concomitant diseases and MAC, so optimizing patients’ therapy seems to be a reasonable and effective approach to limit the development and progression of MAC. Further studies are needed to obtain sufficient clinical evidence and provide useful practical guidelines for the preventive and therapeutic management of MAC.

## Figures and Tables

**Figure 1 ijms-24-03132-f001:**
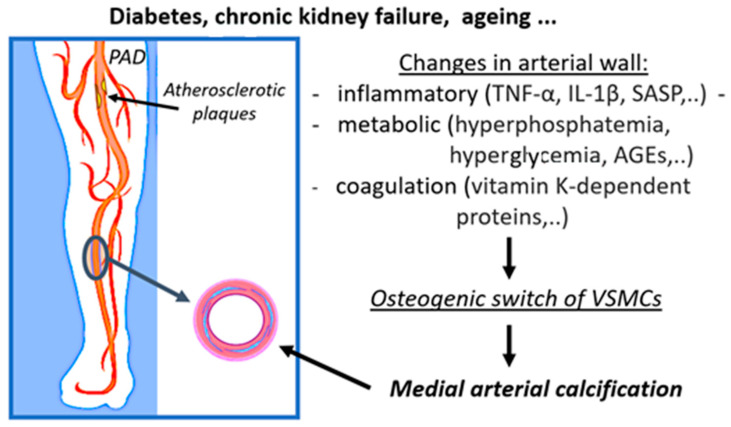
Schematic presentation of the involvement of inflammatory, metabolic, and coagulation processes in the development of medial arterial calcifications in peripheral arterial disease. (PAD: peripheral arterial disease, TNF-α: tumor necrosis factor alpha, IL-1β: interleukin-1beta, SASP: senescence-associated secretory phenotype, AGEs: advanced glycation end products, VSMCs: vascular smooth muscle cells). This figure is adapted from the paper: https://www.mdpi.com/1422-0067/23/19/12054 (accessed on 1 February 2023)

**Table 1 ijms-24-03132-t001:** Inflammatory, metabolic, and coagulative factors involved in MAC pathophysiology and possible mechanisms by which they contribute to MAC development and progression. (TNF-α: tumor necrosis factor alpha, IL-1β: interleukin-1beta, TNAP: tissue-nonspecific alkaline phosphatase, SASP: senescence-associated secretory phenotype, IL-6: interleukin-6, AGEs: advanced glycation end products, VSMCs: vascular smooth muscle cells, ECM: extracellular matrix, MMPs: matrix metalloproteinases, FGF23: fibroblast growth factor 23, LDL: low-density lipoproteins, HDL: high-density lipoproteins, TGF-β:transforming growth factor β).

Factors		Suggested Mechanism	Reference
**Inflammatory Factors**			
TNF-α	Modulates the expression of genes involved in VSMC transformation to osteoblast-like cells	Activation of the cAMP pathway	[36]
	Modulates the expression of genes involved in ECM formation and mineralization		[36]
IL-1 β	Increases the expression of TNAP in VSMCs	Decreases extracellular level of pyrophosphate ions	[37]
SASP development	Increased production of growth factors, proteases, and inflammatory cytokines with osteo-inductive and pro-inflammatory characteristics (Bmp-2, Il-1β, and Il-6)		[29,38,39]
**Metabolic Factors**			
Hyperphosphatemia			
	Promotes precipitation of hydroxyapatite crystals		[40,41]
	Modulation of phenotypic transformation of VSMCs to osteoblast-like cells		[42,43,44,45,46,47,48]
	Extracellular matrix remodeling: - Increased ECM degradation by increasing MMPs and cysteine protease expression, - Increased collagen synthesis - Induces the expression of enzymes that regulate collagen crosslinking and organization		[49,50]
	VSMC apoptosis induction	Disrupts mitochondrial energy metabolism	[44]
	Inhibition of monocyte/macrophage progenitor differentiation into osteoclast-like cells.	Down-regulation of RANKL-induced JNK Akt and NF-κB activation pathways	[51,52,53]
	Modulation of FGF23 levels and Klotho expression	Regulation of serum phosphate levels, vitamin D metabolism, and secondary hyperparathyroidism	[54,55,56,57,58]
Hyperglycemia			
	Increased oxidative stress	Activation of the polyol pathway	[59]
	Inflammatory response induced by advanced glycation end-products (AGEs) - Increased inflammation and oxidative stress - Increased glycation of LDL, HDL, - Activation of the pro-inflammatory inducible nitric oxide (NO)-synthase (iNOS) and decreased NO availability - Increased synthesis of cytokines (IGF-1, PDGF) - Osteogenic differentiation of VSMCs by activating the Notch/Msx2 signaling pathway - Elastin–AGEs crosslinking increases binding of calcium ions to elastin and increased elastin stiffness		[60,61,62,63,64]
	Decreased level of nitric oxide and increased level of endothelin-1	Activation of the protein kinase-C pathway	[59]
	Increased levels of plasminogen activator inhibitor-1 and TGF-β	Stimulation of the hexosamine pathway	[59]
**Coagulative Factors**			
	Stasis of the blood		
	Subendothelial hyperplasia of the intima		[1]
	Impact on coagulation processes	Biological activity of some of the most important proteins involved in MAC pathophysiology is highly dependent on their carboxylation, the process that requires vitamin K as a cofactor	[64,65]

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
