# Peer review of "Inflammatory, Metabolic, and Coagulation Effects on Medial Arterial Calcification in Patients with Peripheral Arterial Disease"

_ijms, 2023, doi:10.3390/ijms24043132_

Round 1
Reviewer 1 Report
Authors present a very detailed review on inflammatory, metabolic and coagulation effects on medial arterial calcification in patients with peripheral arterial disease. This is a very relevant clinical topic and the authors should be congratulated on their work in putting together this manuscript.
1. Introduction (lines 28-30): authors mention hydroxyapatite crystal deposition as one of the "first known features of atherosclerosis". Does this mean that calcification was historically one of the earliest discovered features of the atherosclerotic process, or one of the earliest manifestations of atherosclerosis? Authors do expand on this later in the text (lines 50, 51), but the initial statement may be confusing for readers and authors are requested to clarify this.
2. Lines 74-78: Authors present a good overview on the effects of MAC on peripheral circulation and hemodynamics. If you could please comment on corresponding changes to ankle brachial index (ABI) and changes to the spectral doppler examination of peripheral arteries (loss of triphasic waveform, development of monophasic waveform in late atherosclerosis, etc) that will give readers an excellent idea about clinical correlation of the hemodynamic effects of MAC and atherosclerosis in PAD.
3. Lines 94-99: Please cite BEST-CLI study when commenting on surgical versus endovascular approach to revascularization in PAD.
4. Line 141: Please correct typo. The word "is" has been repeated.
5. Lines 152-154: Please expand on the role of other systemic inflammatory states such as psoriasis, systemic lupus erythematosus and rheumatoid arthritis among others in the development and progression of atherosclerosis, and if at all relevant, MAC.
6. Line 331: grammar incorrect. Please remove the comma after "pathways" and reassess.
7. Line 332: Incorrect use of commas.
8. Line 335: Incorrect use of commas.
9. Line 337: Incorrect use of commas.
10. Please review entire document for appropriateness and uniformity when it comes to using the Oxford comma.
11. Line 369: Why have "dyslipidemia" and "abdominal obesity" been highlighted? Consider removing the underline or expounding on a special relationship between OPG levels and these specific risk factors, if any.
12. Lines 393, 394: Please use full terms for "CT" and "DSA". As these abbreviations have been used <3 times in the manuscript, they do not need to be expanded/defined in the text and full terms will be acceptable.
13. Line 509: Please use full terms for "IVUS" and "OCT". As these abbreviations have been used <3 times in the manuscript, they do not need to be expanded/defined in the text and full terms will be acceptable.
14. Although the focus of this article is the molecular basis of inflammatory, metabolic and coagulation effects on medial arterial calcification, please consider adding some material on the clinical correlation of this topic. This may increase readership, and outlining the clinical implications of such an important problem that affects millions of people worldwide will only increase the relevance of this paper. Some suggestions for clinical additions have been outlined above, but authors are requested to use their discretion and expertise in adding more material as needed.

Author Response
Authors present a very detailed review on inflammatory, metabolic and coagulation effects on medial arterial calcification in patients with peripheral arterial disease. This is a very relevant clinical topic and the authors should be congratulated on their work in putting together this manuscript.
- Introduction (lines 28-30): authors mention hydroxyapatite crystal deposition as one of the "first known features of atherosclerosis". Does this mean that calcification was historically one of the earliest discovered features of the atherosclerotic process, or one of the earliest manifestations of atherosclerosis? Authors do expand on this later in the text (lines 50, 51), but the initial statement may be confusing for readers and authors are requested to clarify this.
Reply: Thank you for this remark. Accumulation of calcium deposits in the vessel wall, in the form of hydroxyapatite with a high degree of crystallization, has been, historically, one of the first recognized features of atherosclerosis and not one of its earliest manifestations. In order to make the introduction more comprehensive, we corrected the initial sentence.
- Lines 74-78: Authors present a good overview on the effects of MAC on peripheral circulation and hemodynamics. If you could please comment on corresponding changes to ankle brachial index (ABI) and changes to the spectral doppler examination of peripheral arteries (loss of triphasic waveform, development of monophasic waveform in late atherosclerosis, etc) that will give readers an excellent idea about clinical correlation of the hemodynamic effects of MAC and atherosclerosis in PAD.
Reply: according to your comment we added a paragraph related to the limitations of ABI use in patients with arterial stiffness as well as changes in Doppler waveform due to PAD progression.
- Lines 94-99: Please cite BEST-CLI study when commenting on surgical versus endovascular approach to revascularization in PAD.
Reply: we cited BEST-CLI study as suggested.
- Line 141: Please correct typo. The word "is" has been repeated.
Reply: the typo was corrected.
- Lines 152-154: Please expand on the role of other systemic inflammatory states such as psoriasis, systemic lupus erythematosus and rheumatoid arthritis among others in the development and progression of atherosclerosis, and if at all relevant, MAC.
Reply: as stated in the article, MAC is not considered predominantly an inflammatory disease, but cytokines could modulate some signalling pathways involved into MAC pathophysiology. To date, the evidence of association between MAC development and progression with any of the inflammatory diseases listed is so scarce that we are not able to provide any insight into the suggested association between MAC and systemic inflammatory diseases.
- Line 331: grammar incorrect. Please remove the comma after "pathways" and reassess.
Reply: the comma after "pathways" was removed and whole sentence reassessed as suggested.
- Line 332: Incorrect use of commas.
Reply: the use of commas was corrected and the whole sentence reassessed.
- Line 335: Incorrect use of commas.
Reply: the use of commas was corrected and the whole sentence reassessed.
- Line 337: Incorrect use of commas.
Reply: the use of commas was corected and the whole sentence reassessed.
- Please review entire document for appropriateness and uniformity when it comes to using the Oxford comma.
Reply: Thank you for this comment. As suggested, the whole document was checked for the uniformity of using the Oxford comma.
- Line 369: Why have "dyslipidemia" and "abdominal obesity" been highlighted? Consider removing the underline or expounding on a special relationship between OPG levels and these specific risk factors, if any.
Reply: In the submitted version of the article those two words were not underlined. We are not sure why this happened in the version that appeared in the pdf format of the text. In the corrected version of the text the underline was removed.
- Lines 393, 394: Please use full terms for "CT" and "DSA". As these abbreviations have been used <3 times in the manuscript, they do not need to be expanded/defined in the text and full terms will be acceptable.
Reply: thank you for this comment. In the corrected version of the article full terms for "CT" and "DSA" were used instead of abbreviations.
- Line 509: Please use full terms for "IVUS" and "OCT". As these abbreviations have been used <3 times in the manuscript, they do not need to be expanded/defined in the text and full terms will be acceptable.
Reply: Reply: thank you for this comment. In the corrected version of the article full terms for "IVUS" and "OCT" were used alongside with the abbreviations.
- 14. Although the focus of this article is the molecular basis of inflammatory, metabolic and coagulation effects on medial arterial calcification, please consider adding some material on the clinical correlation of this topic. This may increase readership, and outlining the clinical implications of such an important problem that affects millions of people worldwide will only increase the relevance of this paper. Some suggestions for clinical additions have been outlined above, but authors are requested to use their discretion and expertise in adding more material as needed.
Reply: the reason that the evidence about clinical implications of MAC in patients with PAD is quite limited lays in the fact that the article is meant for the special issue “Peripheral Artery Disease: Inflammatory, Metabolic and Coagulative Pathogenetic Implications”. The special issue aims to focus on pathophysiological mechanisms and not its clinical relevance. As we fully agree to your comment, we added some information about clinical consequences of MAC in patients with PAD.
Reviewer 2 Report
In this review the authors review the various mechanisms considered to underlie medial arterial calcification in people with peripheral arterial disease. The review is interesting and mostly well-written except a number of minor typos. My only significant concern is about the quality of the figure provided which I believe could better demonstrate the main molecular pathways highlighted later in the review. Otherwise this piece should only need minor correction prior to publication.
Major Comments
Figure 1 is a bit basic and does not really convey any significant information – it would be good to see branches with specific examples of the major metabolic, inflammatory and coagulation pathways involved in MAC from the main text on this diagram. For example demonstration of the effect of TNF-alpha, AGE, and Vitamin K-dependent coagulation factors could be better demonstrated on this figure.
Minor comments
L61 dysfunction of extracellular matrix proteins dysfunctional accumulation and reorganisation of extracellular matrix proteins
L141 Remove repeated “is”
L152-154 - sentence is repetitive. Could shorten “As both CKD and diabetes mellitus are considered inflammatory conditions, inflammation was proposed to be a missing link between those two conditions and the high prevalence of MAC observed in patients with either CKD or diabetes” to ““As both CKD and diabetes mellitus are considered inflammatory conditions, inflammatory signalling was proposed to be a key factor in high prevalence of MAC observed in these patient groups”
L162 Insert “However” before “recent evidence…”
L168-170 Clumsy overly long sentence – please rephrase to something like “ after they are activated by infection, oxidised LDL as well as breakdown products of fibronectin or laminin glycoprotein”
L185 – Where is the level of pyrophosphate decreasing?
L234 – typo – hemodialysis
L343 - Replace “However, the main mechanism,” with “The principal mechanism…”
L373-375: This sentence does not make sense to me as the end of the sentence does not directly relate to the text upstream. Please rephrase. “A recent finding suggests that osteocalcin, the most abundant noncollagenous protein in the bone, is also involved in the regulation of energy metabolism identified bones as an endocrine organ”
L424 Ca2+ instead of Ca+2
Author Response
In this review the authors review the various mechanisms considered to underlie medial arterial calcification in people with peripheral arterial disease. The review is interesting and mostly well-written except a number of minor typos. My only significant concern is about the quality of the figure provided which I believe could better demonstrate the main molecular pathways highlighted later in the review. Otherwise this piece should only need minor correction prior to publication.
Major Comments
Figure 1 is a bit basic and does not really convey any significant information – it would be good to see branches with specific examples of the major metabolic, inflammatory and coagulation pathways involved in MAC from the main text on this diagram. For example demonstration of the effect of TNF-alpha, AGE, and Vitamin K-dependent coagulation factors could be better demonstrated on this figure.
Reply: according to the reviewer’s comment and the information provided in the article the figure was corrected and major pathophysiological mechanisms involved in MAC development and progression are listed. However, according to other reviewer’s comment, in order to make the article more comprehensive we added a separate table with main pathophysiological mechanisms listed. As we considered the table more informative we decided not to added detailed information about main pathophysiological mechanism in the picture.
Minor comments
L61 dysfunction of extracellular matrix proteins dysfunctional accumulation and reorganisation of extracellular matrix proteins
Reply: In the corrected version of the article “dysfunction of extracellular matrix proteins“ was replaced with the suggested sequence.
L141 Remove repeated “is”
Reply: Repetead “is” was removed and whole sentence reassessed.
L152-154 - sentence is repetitive. Could shorten “As both CKD and diabetes mellitus are considered inflammatory conditions, inflammation was proposed to be a missing link between those two conditions and the high prevalence of MAC observed in patients with either CKD or diabetes” to ““As both CKD and diabetes mellitus are considered inflammatory conditions, inflammatory signalling was proposed to be a key factor in high prevalence of MAC observed in these patient groups”
Reply: the sentence was shortened as suggested.
L162 Insert “However” before “recent evidence…”
Reply: The sentence was corrected according to the suggestion.
L168-170 Clumsy overly long sentence – please rephrase to something like “ after they are activated by infection, oxidised LDL as well as breakdown products of fibronectin or laminin glycoprotein”
Reply: Thank you for this comment. The sentence was shorten as follows: “Macrophages are the main source of TNF-α synthesis, after they are activated, by infection, oxidised LDL or ECM breakdown products (e.g. fibronectin or laminin glycoprotein components).”
L185 – Where is the level of pyrophosphate decreasing?
Reply: Thank you for this comment. Pyrophosphate level in the extracellular space decrease as a result of increased expression of TNAP in VSMCs. The remark was included in the corrected version of the article.
L234 – typo – hemodialysis
Reply: The typo was corrected.
L343 - Replace “However, the main mechanism,” with “The principal mechanism…”
Reply: The sentence was corrected according to the suggestion.
L373-375: This sentence does not make sense to me as the end of the sentence does not directly relate to the text upstream. Please rephrase. “A recent finding suggests that osteocalcin, the most abundant noncollagenous protein in the bone, is also involved in the regulation of energy metabolism identified bones as an endocrine organ”
Reply: thank you for this comment. The original sentence was reassessed and rephrased as follows: “A recent finding suggests that osteocalcin, the most abundant noncollagenous protein in the bone, is also involved in the regulation of energy metabolism indicating skeleton as potentially an endocrine organ.”
L424 Ca2+ instead of Ca+2
Reply: Ca+2 was replaced with Ca2+.
Reviewer 3 Report
In this review, the authors present of possible mechanisms of vascular calcification in PAD. comments are below
1. since the authors showed a trendy common mechanism of calcification but it is far away from clinical entities related to PAD in patients. thus, the patients on the title to remove is a relevant way as a suggestion.
2. the authors required categorizing factors related to MAC in a table. to be able to recognize the pathophysiology of calcification in medial calcification.
3. to be able to understand figure 1 for the reader, athero-plaque (delineated white area, next to the leg) should come into the vessel in the illustration. furthermore, the authors suggested transdifferentiation of VSMCs is associated with MAC directly in the figure.
4. However, there is a missing link between Intimal calcification and MAC. Phenotypic switching is often used to describe transdifferentiation of the contractile VSMCs to Intimal calcification as well as Intimal calcification is a feature of plaque in addition, VSMCs produce collagen and elastin, which form a so-called fibrous cap around the lesions. usually, observed as spotty calcifications on the fibrous cap where calcium accumulation is initiated by an increased accumulation of oxLDL in the atherosclerotic lesion contributing to intimal calcification. Therefore the authors required further explanation between MAC and Intimal.
5. Vascular calcification is not only associated with hyperphosphatemia but also hypercalcemia, induces active cell processes through osteogenic differentiation of VSMCS. It is necessary further to delineate.
Author Response
In this review, the authors present of possible mechanisms of vascular calcification in PAD. comments are below
- since the authors showed a trendy common mechanism of calcification but it is far away from clinical entities related to PAD in patients. thus, the patients on the title to remove is a relevant way as a suggestion.
Reply: this article was considered for the special issue “ Peripheral Artery Disease: Inflammatory, Metabolic and Coagulative Pathogenetic Implications”. The title has been chosen on the basis of the issue topic. Considering that, we think that title is appropriate and would like to keep it as it was set at first.
- the authors required categorizing factors related to MAC in a table. to be able to recognize the pathophysiology of calcification in medial calcification.
Reply: in order the make the article more comprehensive and user-friendly we categorized main pathophysiological mechanisms and signalling pathways involved in MAC development and progression in the separate table.
- to be able to understand figure 1 for the reader, athero-plaque (delineated white area, next to the leg) should come into the vessel in the illustration. furthermore, the authors suggested transdifferentiation of VSMCs is associated with MAC directly in the figure.
Reply: the figure was corrected so as to correct the location of the atherosclerotic plaque and put it into the vessel lumen. Due to other reviewer’s comment main pathophysiological mechanisms, explained in the article, were pointed out in the picture.
- However, there is a missing link between Intimal calcification and MAC. Phenotypic switching is often used to describe transdifferentiation of the contractile VSMCs to Intimal calcification as well as Intimal calcification is a feature of plaque in addition, VSMCs produce collagen and elastin, which form a so-called fibrous cap around the lesions. usually, observed as spotty calcifications on the fibrous cap where calcium accumulation is initiated by an increased accumulation of oxLDL in the atherosclerotic lesion contributing to intimal calcification. Therefore the authors required further explanation between MAC and Intimal.
Reply: the focus of the article is on pathophysiologic mechanisms involved in the development and progression of MAC. Atherosclerosis and MAC share some of the most important risk factors and coexist in vast majority of patients with PAD. As stated in the article, those two entities also share some pathophysiologic mechanisms, but their contribution to disease development and progression is different. Atherosclerosis results from subendothelial lipid deposition, macrophage accumulation, smooth muscle cell proliferation, and dysfunctional accumulation and reorganisation of extracellular matrix proteins in response to chronic arterial inflammation. Inflammation and oxidative stress play an important role in atherosclerosis progression, while their contribution to MAC is not considered to be as important.
Intimal calcifications are observed in the vessel wall even before the formation of calcified atherosclerotic plaques, so it was hypothesized that MAC could induce atherosclerotic plaque development and progression by triggering subendothelial hyperplasia. The exact mechanism of their interference has not been fully elucidated. This effect could be paracrine, resulting from phenotypic transformation of both VSMCs and macrophages as well as changes in extracellular matrix of the vessel wall that starts in the media and expand towards intima.
- Vascular calcification is not only associated with hyperphosphatemia but also hypercalcemia, induces active cell processes through osteogenic differentiation of VSMCS. It is necessary further to delineate.
Reply: we agree that increased serum calcium level represents an important factor for the development and progression of MAC. However, the exact mechanism is not explained in detail as for the role of increased serum phosphate levels. According to your suggestion, we added a paragraph about importance of an increased serum calcium levels for the development and progression of MAC.